# Research

analytical chemistry/environmental chemistry/green chemistry

flocculant, dye wastewater, acrylamide polymer, adsorption

**Authors for correspondence:**
Yanbin Wang
e-mail: ybwang@126.com
Zhenhua Li
e-mail: lizhh02006@163.com

This article has been edited by the Royal Society of Chemistry, including the commissioning, peer review process and editorial aspects up to the point of acceptance.

# Synthesis and characterization of cyclodextrin-based acrylamide polymer flocculant for adsorbing water-soluble dyes in dye wastewater

Qiong Su[1,2], Yuxing Wang[1,2], Wanhong Sun[2,3], Junxi Liang[1,2], Shujuan Meng[1,2], Yanbin Wang[1,2] and Zhenhua Li[1,2]

[1]College of Chemical Engineering, Northwest Minzu University, Lanzhou 730030, People's Republic of China
[2]Key Laboratory of Environmental Friendly Composite Materials and Biomass Utility in Universities of Gansu Province, Northwest Minzu University, Lanzhou 730030, People's Republic of China
[3]Department of Experimental Teaching, Northwest Minzu University, Lanzhou 730030, People's Republic of China

ZL, 0000-0001-5403-2364

A novel hydrophobic and cationic cyclodextrin-based acrylamide flocculant (AM-β-CD-DMDAAC) was prepared by chemical oxidative polymerization to adsorb water-soluble dyes in dye wastewater. Fourier transform infrared spectroscopy, X-ray diffraction, scanning electron microscope and thermogravimetric (TG) measurements results demonstrated that the AM-β-CD-DMDAAC was successfully synthesized. The effects of pH, contact time, initial dye concentration, temperature and adsorbent dose on dye removal efficiency for AM-β-CD-DMDAAC flocculants were investigated. The kinetic data were found to follow the pseudo-second-order kinetic model. The equilibrium adsorption data were fitted to the Langmuir isotherm model, with the maximum adsorption capacity of 147.1 mg g$^{-1}$. The adsorbent retained about 60% of the adsorption efficiency after three adsorption/desorption cycles, which implied a promising application as the dye adsorbent.

## 1. Introduction

Dyeing wastewater with deep colour and high COD value seriously pollutes the water environment. The composition of

the discharge effluents and the complexity of contaminants in dyeing wastewater make it one of the hardest types of wastewater to treat, and despite the development and application of various treatment methods, there is still much progress to be made [1–3]. The treatment of dyeing wastewater basically proceeds as follows [4]: (i) pretreatment and primary treatment, wastewater is directed towards the removal of pollutants with the least effort, suspended solids are removed by neutralization, sedimentation, filtration and handled as concentrated solids; (ii) secondary treatment organically combines the biological method and chemical method (mainly flocculation) to remove organic matter and suspended substances; (iii) tertiary treatment uses adsorption, chemical oxidation, ion exchange and membrane separation to remove organic and inorganic pollutants that are difficult to degrade, which is an advanced treatment and is expensive. After secondary treatment, the sewage can generally meet the discharge standard, so flocculation has become an important wastewater treatment method; polyacrylamide and its water-soluble derived copolymers have become more and more widely used in flocculation water treatment in recent years.

Compared with synthetic polymer flocculants, natural polymer flocculants, such as starch, cellulose, chitosan, vegetable gum, protein and microorganisms, have the advantages of being renewable, non-toxic and biodegradable [5–7]. However, good flocculants should also have properties such as large adsorption capacity, acid and alkali resistance, high temperature resistance, short equilibrium time, etc. Since natural materials generally cannot meet such properties, their flocculation efficiency is reduced, and drag resistance and flocculation are easily caused at relatively high concentrations. The natural macromolecule flocculation material can be chemically modified, for example, the cationization is of great significance for improving flocculation properties [8,9].

β-Cyclodextrins (β-CDs) are abundant in nature and have biodegradable and environmentally friendly properties, can be modified by chemical method [10,11], while at the same time, their hydrophobic cavity has a certain inclusion property for organic molecules with moderate radius, and can form inclusion compounds with aromatic compounds through host–guest recognition, so they have high removal efficiency for aromatic compounds in aromatic dye wastewater. It was reported that [12–17], β-CDs combined with traditional flocculating materials such as polyacrylamide can prepare new flocculating materials with high selectivity and high absorption efficiency for organic pollutants. However, in these related references, harmful organic reagents such as polyacrylamide are used in large quantities, which cannot be considered environmentally friendly. How to reduce the use of harmful reagents is an urgent problem to be solved.

In this work, β-CD was used as the main material to synthesize a highly efficient hydrophobic composite polymer flocculant, and throughout the course of the study the amount of harmful reagents used was strictly controlled to protect the environment. Hydrophobic β-CD was introduced into the traditional polyacrylamide flocculating materials, simultaneously based on redox-free radical solution reaction, copolymerized with dimethyl diallylammonium chloride monomer to prepare cationic polyacrylamide–cyclodextrin composite polymer flocculation material to improve flocculation performance [18–21], reduce the accumulation of dyes in the environment and achieve win–win results for economic and environmental benefits.

# 2. Experiment

## 2.1. Materials

β-CD was purchased from Shanghai Shanpu Chemical Co., Ltd. (China) and used after secondary recrystallization. Tosylsulfonyl chloride (TsCl) was from Tianjin Kaixin Chemical Industry Co., Ltd (China). Ethylenediamine (EDA) was purchased from Tianjin Zhiyuan Chemical Reagent Co., Ltd (China). Acryloyl chloride, dimethyl diallyl ammonium chloride (DMDAAC) and methyl acryloxy propyl tris (trimethylsiloxy) silane were purchased from Aladdin reagent (China) and used without treatment. All chemicals were of reagent grade.

## 2.2. Synthesis of acrylamide cyclodextrin

Sulfonylation [22,23]: β-CD (5.0 g) was dissolved in an excess of sodium hydroxide aqueous solution, cooled in an ice water bath. Then, p-toluenesulfonyl chloride (2.0 g) was slowly added into the above solution in batches. The mixture was stirred at 0–5°C for 5 h and filtered, the pH value of the filtrate solution was adjusted to neutrality with 10% hydrochloric acid solution and the filtrate was

**Scheme 1.** Synthesis of AM-β-CD-DMDAAC flocculant.

refrigerated overnight. The crude product was recrystallized from water and dried at 60°C in vacuum for 24 h to obtain a pure white solid of TsO-β-CD (yield: 53.8%).

EDA alkylation [22,23]: The above synthesized TsO-β-CD (5.0 g) and excess EDA (n(TsO-β-CD): n(EDA) = 1 : 1200) were mixed and refluxed at 80°C for 48 h. The unreacted EDA was removed by vacuum distillation, and the residue was treated by acetone. The obtained white precipitation was filtered and purified with methanol, and dried at 60°C in vacuum (yield of EDA-β-CD: 25.7%).

Alkenylation: EDA-β-CD (2.0 g) and sodium bicarbonate (1.0 g) were dissolved in 12 ml of methanol–water mixed solution (the volume of methanol to water = 1 : 2), and stirred for 2 h in an ice bath under $N_2$ protection. Then 2.0 ml of acryloyl chloride was dissolved in 8.0 ml of tetrahydrofuran, and added dropwise into the above mixture solution and reacted for 8 h at room temperature. The solvents were removed under reduced pressure distillation to obtain a pale yellow condensate (gel), which was dissolved by methanol. Then it was precipitated by acetone, filtered, purified with methanol and acetone. The light yellow powders were dried at 60°C in vacuum to obtain the AM-β-CD (yield: 40%).

## 2.3. Synthesis of cyclodextrin-based acrylamide polymer flocculant

The acrylamide cyclodextrin and dimethyl diallyl ammonium chloride (m(AM-β-CD) : m(DMDAAC) = 1 : 3) were dissolved in water, two monomers were 20% of the total mass. Then the mixture of ammonium persulfate and sodium bisulfite (n(APS):n(SBS) = 1 : 1) were added into the above solution in the protection of $N_2$, the mass of the initiators was 0.5% of the weight of the monomer. The mixture was copolymerized for 20 h at 40°C, then cooled to room temperature and poured into acetone to gain the white solid, which was filtered, washed with acetone and dried at 60°C in vacuum for 48 h to obtain the AM-β-CD-DMDAAC flocculant. The synthesis route of AM-β-CD-DMDAAC flocculant is described in scheme 1. The cationic degree of flocculant was determined by chemical titration (yield: 58.3%, cationic degree: 0.85).

## 2.4. Measurement

Fourier transform infrared spectroscopy (FT-IR) spectra of all samples in KBr pellets were recorded using a Nicolet 380 FT-IR spectrophotometer. X-ray diffraction (XRD) studies were performed by using a X'Pert Pro X-ray diffractometer (PANalytical) with Cu Kα as the radiation source. A scanning electron microscope (SEM) analysis was performed using a JSM-6330F (Hitachi, Japan, JEOL) SEM instrument. Prior to the examination, the specimens were coated with a very thin layer of gold. A Thermogravimetric Analyzer (TGA1, Netzsch, Germany) was used to analyse the mass change with temperature at 40–600°C.

## 2.5. Sewage test method

Dye wastewater solution 1 (DWS1) used in the study was prepared by dissolving bromocresol green and thymol blue with the same mass into distilled water. Dye wastewater solution 2 (DWS2) used in the study was prepared by dissolving cresol red and xylenol orange with the same mass into distilled water. The maximum absorption wavelengths of DWS1 and DWS2 were 416 and 438 nm, respectively. In order to test the applicable pH range of the AM-β-CD-DMDAAC, the experiments were conducted at different pH (3, 4, 5, 7, 9 and 11) values with different pretreatment pH (3, 4, 5, 7, 9 and 11) values in 600 mg l$^{-1}$ for 30 min at an agitation speed of 240 rpm, and the pH was adjusted using the HCl and NaOH solution, respectively. In order to obtain the optimized recipe, various dosages (i.e. 20, 30, 40, 50, 60, 70 mg l$^{-1}$) of AM-β-CD-DMDAAC flocculant were applied in 10 mg l$^{-1}$ DWS1 and DWS2 at 25°C.

In the isotherm investigation, AM-β-CD-DMDAAC flocculant were added into various DWS1 and DWS2 initial concentrations of 400, 500, 600, 800 and 1000 mg l$^{-1}$, and shaken at 25°C and 35°C, respectively, for 30 min. To obtain the kinetic data, the composites were suspended in the solutions with DWS1 and DWS2 of 50 mg l$^{-1}$, respectively, at various contact times (5–180 min). At the end of predetermined time intervals, the samples were filtered and determined using a UV–Visible Spectrometer at the maximum absorption wavelength of DWS1 and DWS2. The removal rate and adsorption capacity were calculated according to the following equations:

$$C = \left[\frac{C_0 - C_t}{c_0}\right] \times 100\% \tag{2.1}$$

and

$$q_e = \frac{V \times (C_0 - C_e)}{m}. \tag{2.2}$$

$C_0$, $C_t$ and $C_e$ are the initial, time $t$ and the equilibrium concentrations (mg l$^{-1}$) of DWS1 and DWS2, respectively. $q_e$ is the amount of dye adsorbed per specific amount of adsorbent (mg g$^{-1}$). $V$ (l) is the volume and $m$ (g) is the weight of the adsorbent.

# 3. Results and discussion

## 3.1. Structure and morphology

Figure 1 shows the FT-IR spectra of modified CD and CD-based acrylamide flocculant. According to the literature [24,25], the absorption peaks of TsO-β-CD at 1153 and 1356 cm$^{-1}$ are attributed to the symmetrical and asymmetrical stretching vibration of S=O in sulfonate. The absorption peak at 1643 cm$^{-1}$ is C=C stretching vibration of the benzene ring. Figure 1b is the infrared spectrum of EDA-β-CD, the peak at 3400 cm$^{-1}$ in cyclodextrin is split into three groups of 3200, 2920 and 2800 cm$^{-1}$. The narrowed peak is attributed to the N–H stretching vibration characteristic absorption peaks of the primary amino group and secondary amino groups. Compared with TsO-β-CD infrared spectrum, the characteristic absorption of S=O at 1350 cm$^{-1}$ has disappeared, indicating that the toluenesulfonyl group has been replaced with the EDA group. The absorption peaks of AM-β-CD at 3440 cm$^{-1}$ are attributed to the characteristic peak of cyclodextrin. The peaks at 1590 and 1350 cm$^{-1}$ are due to the N–H and C–N vibration of amide. The peak at 1630 cm$^{-1}$ is due to the C=C and C=O stretching vibration. The appearance of the peaks indicated that the synthesized products contained vinyl double bond, amino group, ester group and cyclodextrin structure. So, acrylamide cyclodextrin was successfully synthesized. The plot d is the infrared spectrum of the AM-β-CD-DMDAAC flocculant. Compared with the curve c, except that the same peak appears at 3440, 1590 and 1350 cm$^{-1}$, the new peak at 2780 cm$^{-1}$ is the intrinsic vibration of the methylene bond, and the peak at 771 cm$^{-1}$ is attributed to =CH$_2$ bending vibration, indicating that the acrylamide cyclodextrin-free radically copolymerized with the cationic monomer DMDAAC.

XRD is one of the most effective methods to analyse the microstructure of crystal materials [26,27]; so, the crystallinity of the modified CD and CD-based acrylamide flocculant were examined by XRD measurements. As shown in figure 2b, sulfonylation weakens the strength of hydrogen bonds in β-CD (figure 2a), leading to a decrease in crystallinity [22]. However, compared with β-CD, TsO-β-CD only shows dispersed crystallization peaks of around 12° and 20°. The peaks of TsO-β-CD become wider

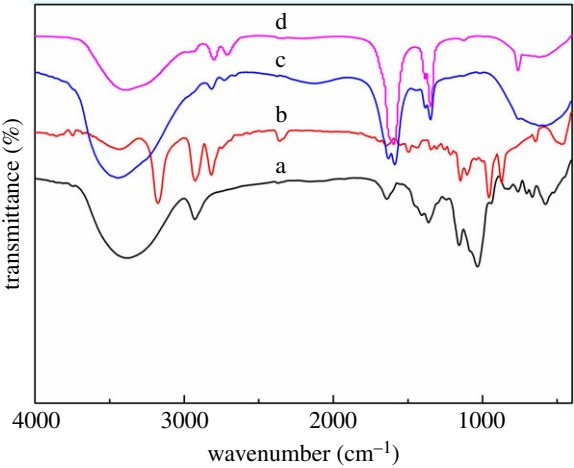

**Figure 1.** FT-IR spectra of TsO-β-CD (a), EDA-β-CD (b), AM-β-CD (c) and AM-β-CD-DMDAAC (d).

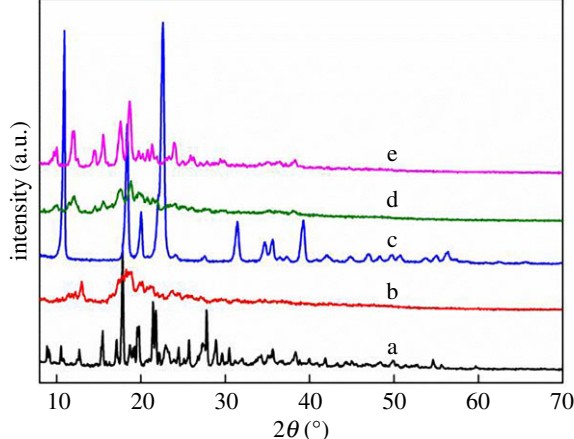

**Figure 2.** XRD of β-CD (a), TsO-β-CD (b), EDA-β-CD (c), AM-β-CD (d) and AM-β-CD-DMDAAC (e).

and the peak intensity decreases obviously. Notably, EDA modification has a great influence on the crystallinity of TsO-β-CD. The strong crystal diffraction peaks of EDA-β-CD were observed at $2\theta$ values of 10°, 18° and 23° (figure 2c). Because acrylamidation, polymerization, cationization and the formation of porous network structure might cause a great change in the crystalline state, the AM-β-CD and AM-β-CD-DMDAAC flocculant only show a relatively broad dispersion crystallization peak of around 10–30°.

SEM was conducted to characterize the as-received pure β-CD, TsO-β-CD, EDA-β-CD, AM-β-CD and AM-β-CD-DMDAAC flocculant, because it is a highly versatile methodology for 2D and 3D materials characterization [28]. The representative SEM images are shown in figure 3. It can be seen that the pure β-CD (figure 3a) prepared shows the regular tetragonal crystal. The morphology of TsO-β-CD, as shown in figure 3b, is microspherical particles of about 10–200 nm in size. Sulfonylation might have severely weakened and destroyed the strength of hydrogen bond in β-CD, which significantly decreased its crystallinity and increased its irregularity. The SEM image of EDA-β-CD is like a blooming flower with of about 10–50 nm in size (figure 3c), indicating larger changes in the crystalline state through EDA. The cyclodextrin-based acrylamide flocculant prepared by redox polymerization has a three-dimensional network structure, which might give it more loose structure, larger gap, better dispersibility and greatly increase the surface area and adsorption sites [14]. It is reported that β-CD-polymer may easily swell in water because the β-CD has some unreacted hydrophilic groups [29]. So, the structure of AM-β-CD-DMDAAC flocculant is beneficial for the adsorption of dye suspended particles of the wastewater.

The thermal stability of materials is an important factor affecting the application [30]; figure 4 shows the thermal stability of TsO-β-CD (a), EDA-β-CD (b), AM-β-CD (c) and AM-β-CD-DMDAAC (d). The initial temperature of weightlessness of TsO-β-CD (curve a) is 190°C. However, the weight loss mainly

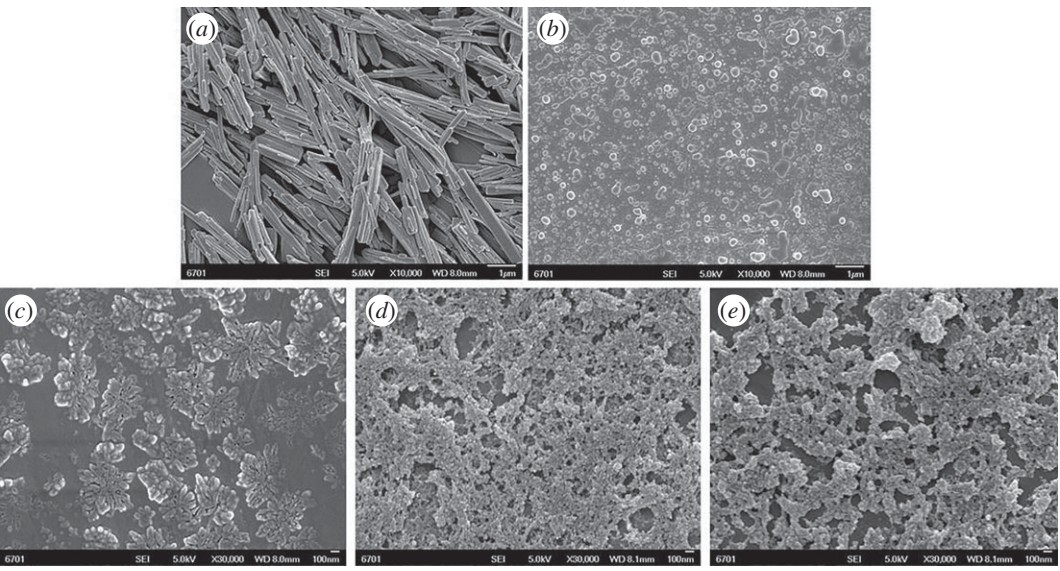

**Figure 3.** SEM of β-CD (*a*), TsO-β-CD (*b*), EDA-β-CD (*c*), AM-β-CD (*d*) and AM-β-CD-DMDAAC flocculant (*e*).

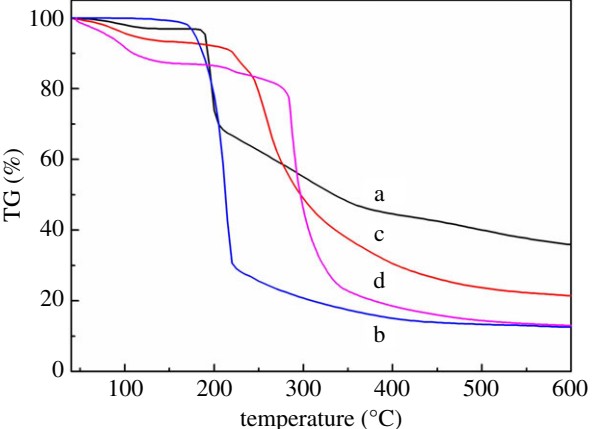

**Figure 4.** TG curves of TsO-β-CD (a), EDA-β-CD (b), AM-β-CD (c) and AM-β-CD-DMDAAC (d).

occurred at 200–400°C, followed by a fast weight loss, and finally the residual mass is about 35%. It indicated that sulfonylation has been effectively performed. The thermal decomposition of EDA-β-CD is continuous and begins to lose weight at about 175°C. The loss of weight reaches 85%. The curve of AM-β-CD exhibited about 70% weight loss in the temperature scale of 220–600°C, which is mainly due to the breaking of the carbon chain. The curve of the AM-β-CD-DMDAAC flocculant has two major changes in the thermal decomposition process due to the fact that the flocculant contains two different structural units: β-CD and DMDAAC. The initial thermal decomposition temperature is obviously reduced to about 100°C, which may be due to the destruction of crystals during the modification of β-CD, but the weight loss is relatively slow, only about 10% at about 280°C. The major weight loss occurs at 300°C and the weight loss reaches 60%. Therefore, the flocculant has good thermal stability in the actual flocculation process.

## 3.2. Adsorption performance of AM-β-CD-DMDAAC

### 3.2.1. Effect of pH and contact time on removal

The pH of the solution plays an important role during the flocculation process. The pH values affect the surface charge of AM-β-CD-DMDAAC flocculant, the degree of ionization and the speciation of flocculation [8]. To explore the influence of the solution pH on the removal, flocculation experiments were conducted over a pH range between 3.0 and 11.0. The removal rate of DWS1 and DWS2 was

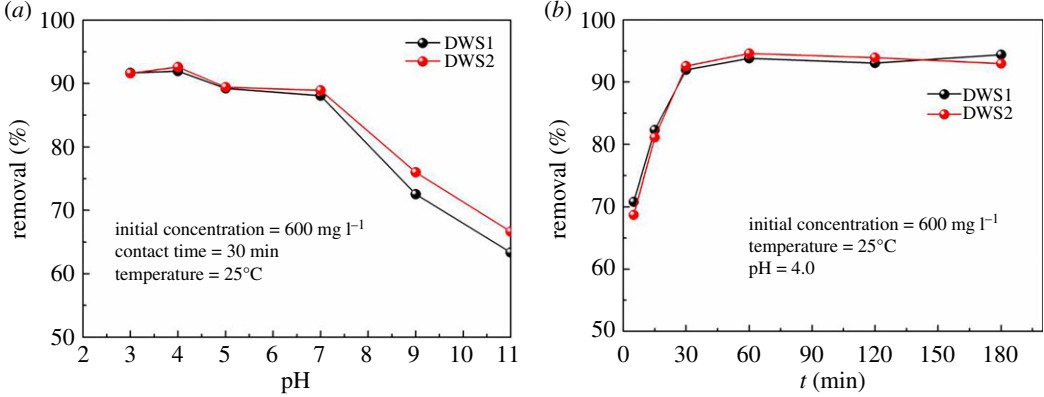

**Figure 5.** Effect of pH (*a*) and contact time (*b*) on removal of AM-β-CD-DMDAAC.

highly pH dependency. As shown in figure 5*a*, when the pH values of the solution were in the range of 3.0–7.0, the removal of DWS1 and DWS2 was high. The AM-β-CD-DMDAAC exhibited maximum adsorption capacity at pH 4. This variation tendency is consistent with a previous report [3]. However, when the pH of the solution was higher than 7.0, the removal of DWS1 and DWS2 reduced sharply with the increase in pH. The removal efficiency of DWS1 and DWS2 was similar. The reason might be that the removal of anionic dye is directly affected by positive charge intensity. However, the intensity of positive charge is mainly determined by the cationic and conformation of AM-β-CD-DMDAAC flocculant [13].

The effect of oscillation time on removal of DWS1 and DWS2 is shown in figure 5*b*. The results revealed that the removal of DWS1 and DWS2 was rapidly increased with the increase in interaction time and then reached an adsorption equilibrium in about 30 min. In just 5 minutes, the removal rate of DWS1 and DWS2 had reached 70.82% and 68.72%, respectively. The removal of DWS1 and DWS2 was about 91.94% and 92.58% after 30 min of the reaction, respectively. The maximum removal is 93.81% and 94.59% for DWS1 and DWS2 after 60 min, respectively. Thus, the adsorption reached equilibrium within half an hour. The fast sedimentation showed that this process was controlled by the bridging effect of charge neutralization [31].

### 3.2.2. Effect of initial concentration, temperature, dosage on removal

The effect of the initial concentration of DWS1 and DWS2 on removal is shown in figure 6*a*. The removal decreased slightly with the increase in the initial concentration of DWS1 and DWS2 range of 400–600 mg l$^{-1}$. The removal was about 91.94% and 92.58% for 600 mg l$^{-1}$ of DWS1 and DWS2, respectively. But, when the concentration of DWS1 and DWS2 goes beyond a certain range, the removal of AM-β-CD-DMDAAC flocculants shows a obviously decreasing trend. When the dye concentration was higher than 800 mg l$^{-1}$, the removal of DWS1 and DWS2 dropped rapidly. It may be that the already formed precipitates were encapsulated by the ions in the solution, thereby preventing the dye molecules in the solution from being further adsorbed by the adsorbent [32].

The flocculant had high removal efficiency to DWS1, especially for a concentration of DWS1 lower than 500 mg l$^{-1}$ (figure 6*b*). The removal rate was as high as 99.89% at 35°C and 99.54% at 25°C for 400 mg l$^{-1}$ of DWS1. However, for 1000 mg l$^{-1}$ of DWS1, the removal rate of DWS1 could still reach 78.62% at 35°C and 71.62% at 25°C. Compared with removal efficiency under different temperature, it could be clearly observed that the removal at 35°C was slightly higher than that at 25°C. With the increase in temperature, the speed of molecules and the probability of collision between DWS1 and AM-β-CD-DMDAAC was improved, which increased adsorption opportunity and enhanced the bridging adsorption effect on long-chain AM-β-CD-DMDAAC [33].

A good adsorbent should be able to remove a relatively high amount of adsorbate at lower doses, which not only increases the adsorption capacity of the adsorbent but also reduces the overall cost of the adsorption process [34]. Figure 6*c* shows the effect of AM-β-CD-DMDAAC dosage on the adsorption of DWS1 and DWS2. The effect of the AM-β-CD-DMDAAC dose on the adsorption of DWS1 and DWS2 was studied by varying the mass of the AM-β-CD-DMDAAC in the range of 20–70 mg. It was observed that the removal increased from 77.57 to 91.94% for DWS1 and from 77.82 to 90.55% for DWS2 with an increase in flocculant dose from 20 to 50 mg. However, the removal decreased with the increase in the dosage of flocculant from 50 to 70 mg. That is to say, the

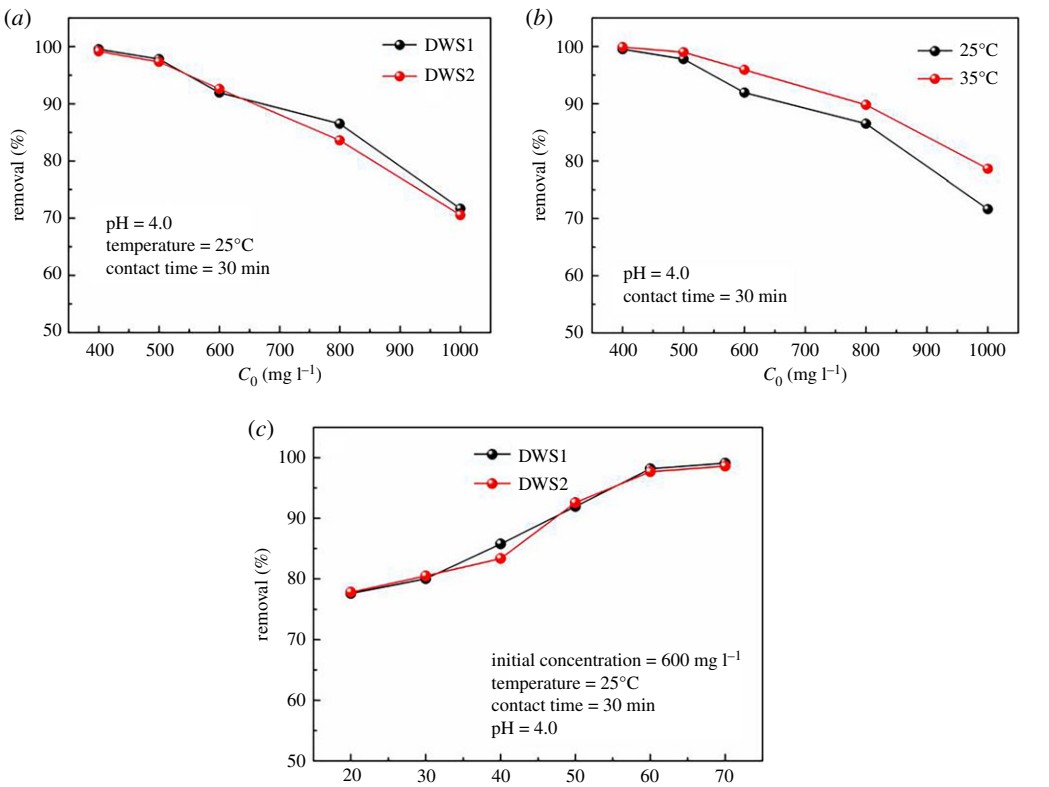

**Figure 6.** Effect of initial concentration of DWS1 and DWS2 (*a*), temperature (*b*) and flocculant dosage (*c*) on removal of AM-β-CD-DMDAAC.

flocculation process has the optimum flocculation concentration (OFC) in a certain dosage range of flocculants. When the concentration of flocculant is lower than OFC, flocculant cannot effectively neutralize the charges on the surface of dye, and there is still a large electrostatic repulsion force between dye anions, so the pollutants cannot accumulate and precipitate to form flocs. When flocculant concentration is higher than the OFC, the excess flocculant will form clathrate compound on the surface of dye molecules or formed flocs, which reduces the bridging effect between the flocculant and dye anions, resulting in weakening of the flocculation effect. At the same time, a large number of opposite charges are adsorbed on the surface of the pollutant particles, which causes the electrostatic repulsion of pollutant particles to increase again, and the flocculation effect decreases [31,35,36].

### 3.2.3. Flocculation mechanism

The flocculation process involves many factors and is the effect of their synergistic interaction. The flocculation process may be relatively complicated. The removal of dye involves the physical and chemical interaction between flocculants and dye molecules surfaces, such as electric neutralization, bridging effect, subject and object recognition, and coil sweeping and net catching, etc. [37,38]. The surface of pollutants in the dye wastewater carries the same type of charge, and there is a great electrostatic repulsive force between the pollutant particles, which could not accumulate and precipitate. When flocculants with opposite electrical properties are added, the charges on the surface of the pollutants will be partially neutralized, the electrostatic repulsion between pollutant particles in the wastewater could be effectively reduced, and the pollutant particles might be accumulated and precipitated to form flocs. Figure 7 shows an electrostatic interaction between the sulfonic acid group of the dye and the quaternary ammonium salt of the flocculant. This electrostatic interaction would depend on the number of sulfonic acid groups in the dye molecule and cationic degree of the flocculant. Hydrophilic groups -NH in acrylamide-based cyclodextrin flocculants can also form stable coordination compounds with having unshared pair electrons of dye molecules. In the process of flocculation, when the active functional groups in the polymer flocculant contact with the pollutant particles, the adsorption occurs and the bridge-building aggregates to form large flocs, and then precipitates [39]. Only with more active functional groups and a more branched chain of flocculant

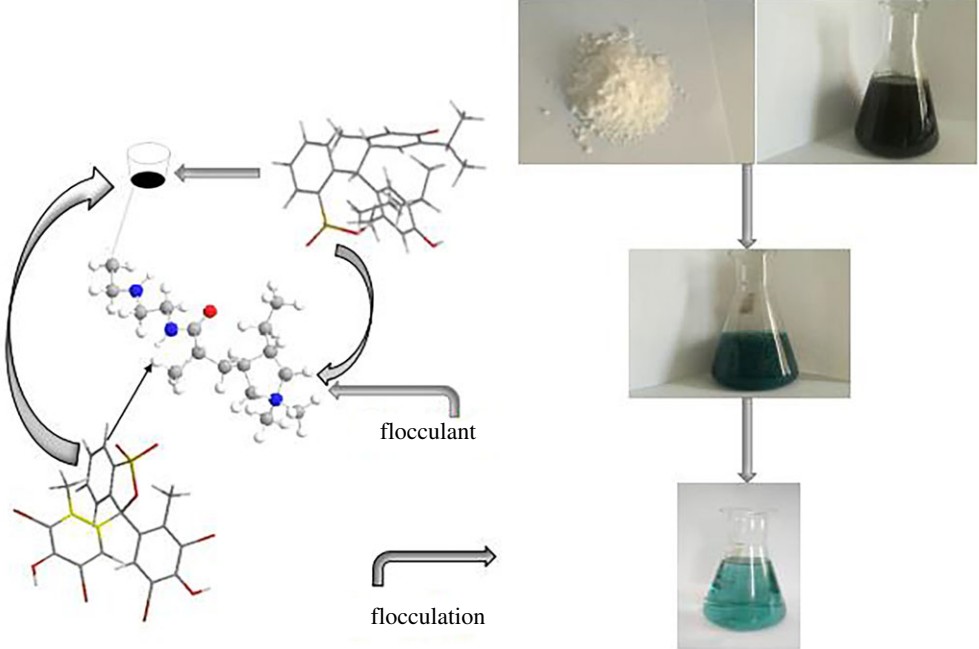

**Figure 7.** The speculation of the possible flocculation mechanism.

can we see more obvious bridging effect and better adsorption efficiency. The large size net-like structure flocs formed by electric neutralization and bridging adsorption, which have a large surface area, many active sites and strong adsorption capacity, and can quickly and effectively roll sweep and net catch the micropollutants that have not yet formed flocs in sewage. This serves to make it aggregate, precipitate and improve the efficiency of sewage treatment [40]. The β-CD units of acrylamide cyclodextrin flocculant have hydrophobic cavities and can selectively adsorb hydrophobic aromatic dye molecules through host–guest recognition to improve flocculation and adsorption efficiency. The host–guest identification between flocculant and aryl group may be enhanced with the increase in the number of the dye aromatic rings.

### 3.2.4. Flocculation kinetics

Kinetic studies were carried out at 25°C with initial concentrations of DWS1 (600 mg l$^{-1}$) and DWS2 (600 mg l$^{-1}$) at various contact times (5–180 min). The pseudo-first-order model relating to the physical diffusion, and pseudo-second-order model used to determine whether the adsorption rate is controlled by chemical adsorption mechanism, are applied extensively to analyse the kinetic data [41]. The pseudo-first-order model is arranged and described as follows:

$$\log(q_e - q_t) = \log q_e - \left(\frac{k_1}{2.303}\right) \times t, \tag{3.1}$$

where $t$ (min) is the adsorption time, $q_t$ (mg g$^{-1}$) is the adsorption amount at some time $t$, $k_1$ (min$^{-1}$) is a constant of the diffusion rate, and $q_e$ and $k_1$ can be obtained from the fitting equation by analysing the plot of $\log(q_e - q_t)$ versus $t$. The pseudo-second-order model can be arranged and described as follows:

$$\frac{t}{q_t} = \frac{1}{k_2 q_e^2} + \frac{t}{q_e}, \tag{3.2}$$

where $k_2$ (g (mg min)$^{-1}$) is the rate constant of the pseudo-second-order equation. $q_e$ and $k_2$ can be obtained from the fitting equation by analysing the plot of $t/q_t$ versus $t$.

Figure 8$a$ shows a plot of $\log(q_e - q_t)$ versus $t$ for the pseudo-first-order equation. Figure 8$b$ shows a plot of $t/q_t$ versus $t$ for the pseudo-second-order equation. The correlation coefficients for the pseudo-first-order equation obtained were low, but the correlation coefficient was 0.9999 and 0.9997 for the pseudo-second-order equation, and the calculated adsorption capacity from the pseudo-second-order equation was 113.6 mg g$^{-1}$, which was in accordance with the experimental values of 113.3 and 113.5 mg g$^{-1}$. The results indicated that these kinetic data agreed with the pseudo-second-order equation. The $k_2$ of DWS1 and DWS2 are calculated to be 0.0084 g (mg min)$^{-1}$ and 0.0082 g (mg min)$^{-1}$, respectively.

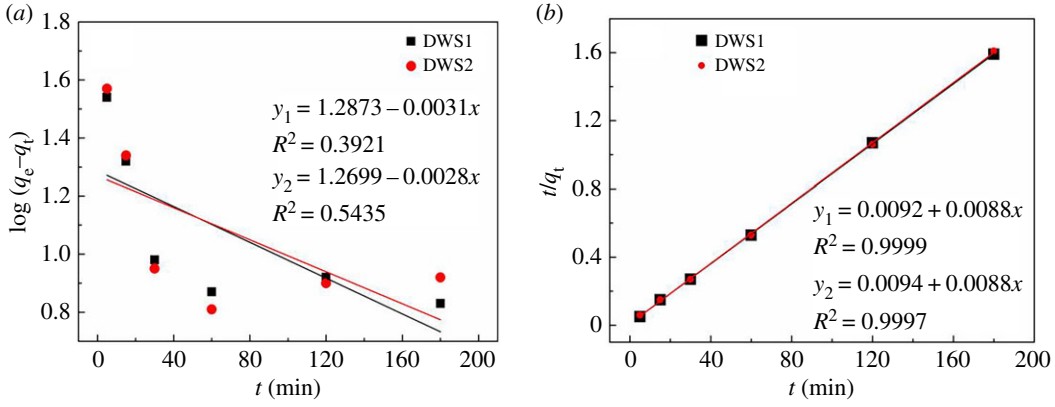

**Figure 8.** Kinetic fitting plots of the pseudo-first-order (*a*) and pseudo-second-order (*b*) equations.

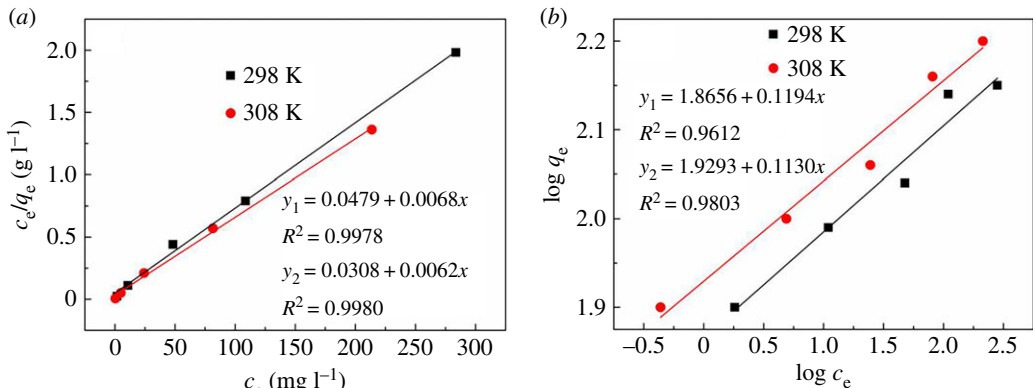

**Figure 9.** Langmuir (*a*) and Freudlich (*b*) isotherms for DWS1 removal by AM-β-CD-DMDAAC.

The Langmuir and Freundlich models were used to fit adsorption behaviour by AM-β-CD-DMDAAC to evaluate the flocculation performance. The Langmuir isotherm and the Freundlich isotherm can be described as formula (3.3) and (3.4), respectively [42]

$$\frac{c_e}{q_e} = \frac{1}{bq_{max}} + \frac{c_e}{q_{max}}, \tag{3.3}$$

where $C_e$ is the equilibrium concentration (mg l$^{-1}$), $q_e$ is the amount of dye molecules adsorbed by flocculant at equilibrium (mg g$^{-1}$), $q_{max}$ is the adsorption capacity of AM-β-CD-DMDAAC (mg g$^{-1}$) and $b$ is a constant (l mg$^{-1}$).

$$\log q_e = \log K_f + \frac{1}{n}\log C_e, \tag{3.4}$$

where $q_e$ is the amount of dye molecules adsorbed on the surface of AM-β-CD-DMDAAC at equilibrium (mg g$^{-1}$), $C_e$ is the equilibrium concentration, and $K_f$ and $n$ are constants of the Freundlich model.

The AM-β-CD-DMDAAC flocculants were used to treat DWS1 with concentrations ranging from 400 to 1000 mg l$^{-1}$ at the different temperatures (25 and 35°C). The adsorption isotherms are shown in figure 9*a,b*. The Langmuir and Freundlich adsorption constants and the corresponding correlation coefficients are listed in table 1. The results showed that the adsorption process was more similar to the Langmuir isotherm model than the Freundlich isotherm model.

### 3.2.5. Recycling

To investigate the stability and the potential regeneration of AM-β-CD-DMDAAC flocculant, the experiments of adsorption and desorption of DWS1 and DWS2 were repeated four times. A certain quality of AM-β-CD-DMDAAC flocculant (50 mg) was added into 10 ml of DWS1 and DWS2 solution (600 mg l$^{-1}$) and shaken for 30 min, then the solution was centrifuged, and the concentrations of the target molecules in the solution were determined using an UV–Visible Spectrometer. After each test,

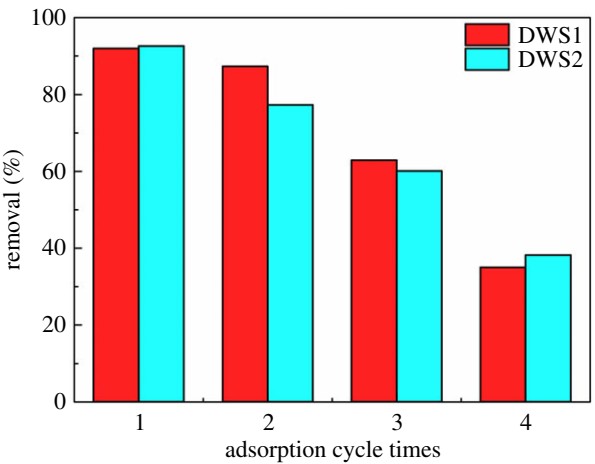

**Figure 10.** Desorption and regeneration of AM-β-CD-DMDAAC flocculant.

**Table 1.** Isotherm parameters for the adsorption of dye molecules on AM-β-CD-DMDAAC.

| $T$ (K) | Langmuir model | | | Freundlich model | | |
|---|---|---|---|---|---|---|
| | $q_{max}$ (mg g$^{-1}$) | $b$ (l mg$^{-1}$) | $R^2$ | $K_f$ | $n$ | $R^2$ |
| 298 | 147.1 | 0.14 | 0.9978 | 73.4 | 8.4 | 0.9612 |
| 308 | 161.3 | 0.20 | 0.9980 | 85.0 | 8.8 | 0.9803 |

AM-β-CD-DMDAAC flocculant was soaked in 0.01 mol l$^{-1}$ NaOH and acetone eluent for 4–8 h, filtered and washed with deionized water and ethanol, then dried under vacuum at 50°C for 24 h, and again subjected to adsorption processes to determine the reusability. The result is shown in figure 10. After four cycles of flocculation experiments, the removals of DWS1 were 91.94%, 87.3%, 62.9% and 35.0%, respectively. The removal efficiency of DWS2 is basically the same as that of DWS1. Although the removal efficiency of DWS1 and DWS2 decreases slowly with the progress of desorption and regeneration, the removal efficiency of DWS1 and DWS2 still stays at a high level. Therefore, the prepared AM-β-CD-DMDAAC is suitable for adsorbent.

## 4. Conclusion

Environmentally friendly and degradable flocculating materials with high selectivity and efficient removal for dye pollutants in wastewater have gradually become the development trend of flocculants. The environmentally-friendly and degradable β-CD was used as raw material, and was chemically modified, then polymerized and ionized with dimethyl diallyl ammonium chloride by aqueous solution polymerization to prepare a novel hydrophobic and strong cationic cyclodextrin-based acrylamide polymer flocculant. Flocculation behaviour of AM-β-CD-DMDAAC flocculants on dye removal was studied at different pH values of dye solution, the contact time, initial dye concentration, temperature and dose of flocculant. The removal rate of DWS1 and DWS2 exceeded 65% in 5 min and almost reached equilibrium in 30 min at 25°C. The optimum flocculation dose OFD was 50 mg. The pH values of the initial solution were favourable at slightly acidic and near neutral conditions (pH 3.0–7.0). The AM-β-CD-DMDAAC flocculant mainly performs flocculation and adsorption of dyestuff wastewater through electric neutralization, adsorption bridging effect and host–guest recognition. The results indicate that the synthesized AM-β-CD-DMDAAC had a higher removal for dyestuff wastewater, and it can be used for flocculation and decoloration of industrial dye wastewater.

Data accessibility. Our data are deposited at the Dryad Digital Repository: https://dx.doi.org/10.5061/dryad.cc2fqz62j [43].
Authors' contributions. Q.S. and Y.W. designed the study. S.M. and J.L. prepared all samples for analysis. W.S. and Y.W. collected and analysed the data. Q.S. and Z.L. interpreted the results and wrote the manuscript. All authors gave final approval for publication.
Competing interests. We declare we have no competing interests.

Funding. This work was supported by the National Natural Science Foundation of China (grant nos 51563022, 21968032), the Fundamental Research Funds for the Central Universities (grant no. 31920190012), the fund of the Double First-class and Characteristic Development Guide of Northwest Minzu University (grant no. 11080316) and the fund of Teaching Quality and Reform Engineering Project of Gansu University (grant nos 2019GSSYJXSFZX-01, 2019GSJXCGPY-16).

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
