## [Reviewer comments · Royal Society Open Science]

Review History

RSOS-191519.R0 (Original submission)

Review form: Reviewer 1

Is the manuscript scientifically sound in its present form?

Yes

Are the interpretations and conclusions justified by the results?

Yes

Is the language acceptable?

Yes

Do you have any ethical concerns with this paper?

No

Have you any concerns about statistical analyses in this paper?

No

Recommendation?

Accept with minor revision (please list in comments)

Comments to the Author(s)

1. add the recycling experimental details, i.e. how to recycle the AM- β -CD-DMDAAC
2. the authors gave the Scheme 1 in the end of manuscript, but did not cite it in the main text
3. why the Ts group react with the -OH group (C6) on the top side of beta CD, not in the bottom side (C2, C3), and how many Ts groups grafted on one beta CD?
4. in the results and discussion part, need to cite more references to support your interpretations and the analyse results (IR,XRD,...)
5. page 4 line 9 "Studies have shown that their flocculation properties can be comparable to that of high-quality synthetic polymer flocculants, but their flocculation efficiency are low", need to specify the flocculation properties, since flocculation efficiency is one of the flocculation properties.

Review form: Reviewer 2**Is the manuscript scientifically sound in its present form?**

No

Are the interpretations and conclusions justified by the results?

Yes

Is the language acceptable?

No

Do you have any ethical concerns with this paper?

No

Have you any concerns about statistical analyses in this paper?

No

Recommendation?

Major revision is needed (please make suggestions in comments)

Comments to the Author(s)

The manuscript titled "Synthesis and characterization of cyclodextrin-based acrylamide polymer flocculant for adsorbing water-soluble dyes in dye wastewater" has reported extensive data related to synthesis, characterization and application. According to the claims by authors it is mentioned that the material prepared was environmentally friendly and degradable. The authors have used β -CD as raw material, and was chemically modified, then polymerized and ionized with dimethyl diallyl ammonium chloride by aqueous solution polymerization to prepare a hydrophobic and cationic cyclodextrin-based acrylamide polymer flocculant. The work reported here lacks novelty. There has been plenty of similar materials reported in literature. The authors need to clearly indicate the novelty of this work before it is considered for publication. In addition the synthesis involves very long hours and this shows the method has no practical applicability due to the high energy cost. According to results it seems to me that this method requires extensive amount of resources which cannot be considered as environmentally friendly. In addition the manuscript needs polishing of the language. The FTIR, XRD and TGA data were not supported with relevant literature. Also some key references are missing in the reference list and need to be corrected.

Decision letter (RSOS-191519.R0)

28-Oct-2019

Dear Miss Li:

Title: Synthesis and characterization of cyclodextrin-based acrylamide polymer flocculant for adsorbing water-soluble dyes in dye wastewater
Manuscript ID: RSOS-191519

The editor assigned to your manuscript has now received comments from reviewers. We would like you to revise your paper in accordance with the referee and Subject Editor suggestions which can be found below (not including confidential reports to the Editor). Please note this decision does not guarantee eventual acceptance.

Please submit your revised paper before 20-Nov-2019. Please note that the revision deadline will expire at 00.00am on this date. If we do not hear from you within this time then it will be assumed that the paper has been withdrawn. In exceptional circumstances, extensions may be possible if agreed with the Editorial Office in advance. We do not allow multiple rounds of revision so we urge you to make every effort to fully address all of the comments at this stage. If deemed necessary by the Editors, your manuscript will be sent back to one or more of the original reviewers for assessment. If the original reviewers are not available we may invite new reviewers.

Please also include the following statements alongside the other end statements. As we cannot publish your manuscript without these end statements included, if you feel that a given heading is not relevant to your paper, please nevertheless include the heading and explicitly state that it is not relevant to your work.

- Acknowledgements

RSC Associate Editor:
Comments to the Author:
(There are no comments.)

RSC Subject Editor:
Comments to the Author:
(There are no comments.)

Reviewers' Comments to Author:
Reviewer: 1

Comments to the Author(s)

1. add the recycling experimental details, i.e. how to recycle the AM- β -CD-DMDAAC
2. the authors gave the Scheme 1 in the end of manuscript, but did not cite it in the main text
3. why the Ts group react with the -OH group (C6) on the top side of beta CD, not in the bottom side (C2, C3), and how many Ts groups grafted on one beta CD?
4. in the results and discussion part, need to cite more references to support your interpretations and the analyse results (IR,XRD,...)
5. page 4 line 9 "Studies have shown that their flocculation properties can be comparable to that of high-quality synthetic polymer flocculants, but their flocculation efficiency are low", need to specify the flocculation properties, since flocculation efficiency is one of the flocculation properties.

Reviewer: 2

Comments to the Author(s)

The manuscript titled "Synthesis and characterization of cyclodextrin-based acrylamide polymer flocculant for adsorbing water-soluble dyes in dye wastewater" has reported extensive data related to synthesis, characterization and application. According to the claims by authors it is mentioned that the material prepared was environmentally friendly and degradable. The authors have used β -CD as raw material, and was chemically modified, then polymerized and ionized with dimethyl diallyl ammonium chloride by aqueous solution polymerization to prepare a hydrophobic and cationic cyclodextrin-based acrylamide polymer flocculant. The work reported here lacks novelty. There has been plenty of similar materials reported in literature. The authors need to clearly indicate the novelty of this work before it is considered for publication. In addition the synthesis involves very long hours and this shows the method has no practical applicability due to the high energy cost. According to results it seems to me that this method requires extensive amount of resources which cannot be considered as environmentally friendly. In addition the manuscript needs polishing of the language. The FTIR, XRD and TGA data were not supported with relevant literature. Also some key references are missing in the reference list and need to be corrected.

Author's Response to Decision Letter for (RSOS-191519.R0)

See Appendix A.

Decision letter (RSOS-191519.R1)

02-Dec-2019

Dear Miss Li:

Title: Synthesis and characterization of cyclodextrin-based acrylamide polymer flocculant for adsorbing water-soluble dyes in dye wastewater

Manuscript ID: RSOS-191519.R1

It is a pleasure to accept your manuscript in its current form for publication in Royal Society Open Science. The chemistry content of Royal Society Open Science is published in collaboration with the Royal Society of Chemistry.

RSC Associate Editor
Comments to the Author:
(There are no comments.)

Reviewer(s)' Comments to Author:

Appendix A

Dear editor:

Thank you very much for your e-mail of Oct. 28, 2019 with regard our manuscript (**RSOS-191519**) together with the comments. The manuscript has been revised carefully according to the opinions of reviewer. The main amendments have been listed in the following:

(Markers according to pages and lines of original manuscript)

Manuscript Number: **RSOS-191519**

Title: **Synthesis and characterization of cyclodextrin-based acrylamide polymer flocculant for adsorbing water-soluble dyes in dye wastewater**

Reviewers' Comments:

Reviewer: 1

Comments to the Author(s)

Q1: Add the recycling experimental details, i.e. how to recycle the AM- β -CD-DMDAAC.

Answer: Thanks for your advice. We have added the experimental details in the revision.

Section 3.2.5, “The regeneration and recycling of AM- β -CD-DMDAAC was studied by four consecutive cycle experiments.” **changed to** “To investigate the stability and the potential regeneration of AM- β -CD-DMDAAC flocculant, filtered and washed with deionized water and ethanol, then dried under vacuum at 50 °C for 24 h, and again subjected to adsorption processes to determine the reusability.”

Q2: The authors gave the Scheme 1 in the end of manuscript, but did not cite it in the main text.

Answer: We are sorry for our mistake.

Section 2.3, we **added** “The synthesis route of AM- β -CD-DMDAAC flocculant is described in Scheme 1.” **before** “The cationic degree of flocculant was determined by chemical titration method. (yield: 58.3%, cationic degree: 0.85).”

Q3: Why the Ts group react with the -OH group (C6) on the top side of beta CD, not in the bottom side (C2, C3), and how many Ts groups grafted on one beta CD?

Answer: According to the literatures [17, 22], C₆ belongs to the primary hydroxyl group, while C₂ and C₃ belong to the secondary hydroxyl group. The primary hydroxyl group has higher reactivity in ice water bath.

In order to determine the number of Ts groups grafted on one beta CD, detailed information on the elemental composition of β -CD-OTs was provided by electron microscopy combined with energy-dispersive X-ray spectroscopy (SEM-EDS) (Figure R1). According to the ratio of C and S, the number of Ts groups grafted on one beta CD can be approximated.

Figure R1 Detailed information on the elemental composition of β -CD-OTs was provided by electron microscopy combined with energy-dispersive X-ray spectroscopy (SEM-EDS).

Q4: In the results and discussion part, need to cite more references to support your interpretations and the analyse results (IR, XRD,)

Answer: According to your advice, we **have supplemented** the references.

Section 3.1, Structure and morphology, paragraph 1 (FT-IR), one reference (ref. 25) was added here.

“According to the literatures [24, 25],” **was added before** “the absorption peaks of TsO- β -CD at 1153 and 1356 cm^{-1} is attributed to the symmetrical and.....”

Section 3.1, Structure and morphology, paragraph 2 (XRD), two references (refs. 26 and 27) were added here.

“The crystallinity of the modified CD and CD-based acrylamide flocculant were examined by XRD measurements.” **changed to** “X-ray diffraction is one of the most effective methods to analyze the microstructure of crystal materials [26,27], so, the

crystallinity of the modified CD and CD-based acrylamide flocculant were examined by XRD measurements.”

Section 3.1, Structure and morphology, **paragraph 3 (SEM)**, one reference (ref. 28) was added here.

“Figure 3 shows the morphological analyses of the as-received pure β -CD, the TsO- β -CD, EDA- β -CD, AM- β -CD and AM- β -CD-DMDAAC flocculant.” **changed to** “SEM was conducted to characterize the as-received pure β -CD, TsO- β -CD, EDA- β -CD, AM- β -CD and AM- β -CD-DMDAAC flocculant, because it is a highly versatile methodologies for 2D and 3D materials characterization [28]. The representative SEM images are shown in Figure 3.”

Section 3.1, Structure and morphology, **paragraph 4 (TG)**, one reference (ref. 30) was added here.

“The thermal stability of materials is an important factor affecting the application [30],” was added before “Figure 4 shows the thermal stability of TsO- β -CD (a), EDA- β -CD (b), AM- β -CD (c) and AM- β -CD-DMDAAC (d). The initial temperature of weightlessness of TsO- β -CD (curve a) is 190 °C.”

Other references’ order also changed correspondingly in revision.

Q5: Page 4 line 9 “Studies have shown that their flocculation properties can be comparable to that of high-quality synthetic polymer flocculants, but their flocculation efficiency are low”, need to specify the flocculation properties, since flocculation efficiency is one of the flocculation properties.

Answer: We are sorry for our mistake. **Section 1, paragraph 2,**

“Natural polymer flocculants such as starch, cellulose, chitosan, vegetable gum, protein and microorganism, etc are renewable, non-toxic and biodegradable [5-7].....and drag resistance and flocculation are easily caused at relatively high concentrations.” **changed to** “Compared with synthetic polymer flocculants, natural polymer flocculants, such as starch, cellulose, chitosan, short equilibrium time etc., since natural materials generally cannot meet such properties, their flocculation efficiency is reduced, and drag resistance and flocculation are easily caused at relatively high concentrations.”

Reviewer: 2

Comments to the Author(s)

The manuscript titled “Synthesis and characterization of cyclodextrin-based acrylamide polymer flocculant for adsorbing water-soluble dyes in dye wastewater” has reported extensive data related to synthesis, characterization and application. According to the claims by authors it is mentioned that the material prepared was environmentally friendly and degradable. The authors have used β -CD as raw material, and was chemically modified, then polymerized and ionized with dimethyl diallyl ammonium chloride by aqueous solution polymerization to prepare a hydrophobic and cationic cyclodextrin-based acrylamide polymer flocculant. The work reported here lacks novelty. There has been plenty of similar materials reported in literature. The authors need to clearly indicate the novelty of this work before it is considered for publication. In addition the synthesis involves very long hours and this shows the method has no practical applicability due to the high energy cost. According to results it seems to me that this method requires extensive amount of resources which cannot be considered as environmentally friendly. In addition the manuscript needs polishing of the language. The FTIR, XRD and TGA data were not supported with relevant literature. Also some key references are missing in the reference list and need to be corrected.

Answer: Thanks for your advices and according to your advice, we **have revised** our paper carefully. The main modification details are as follows.

Q1: The manuscript needs polishing of the language.

Answer: The grammar and spelling mistakes **have been corrected** in revision.

Q2: The FTIR, XRD and TGA data were not supported with relevant literature. Also some key references are missing in the reference list and need to be corrected.

Answer: We are sorry for our mistake. The references listed in manuscript **have been corrected**. Some key references **have been added** in our revision. The modification details of Section 3.1 are as follows, and the changes of other parts are shown in the Section references.

Section 3.1, Structure and morphology, **paragraph 1 (FT-IR)**, one reference (ref. 25) was added here.

“According to the literatures [24, 25],” **was added before** “the absorption peaks of TsO-β-CD at 1153 and 1356 cm⁻¹ is attributed to the symmetrical and.....”

Section 3.1, Structure and morphology, **paragraph 2 (XRD)**, two references (refs. 26 and 27) were added here.

“The crystallinity of the modified CD and CD-based acrylamide flocculant were examined by XRD measurements.” **changed to** “X-ray diffraction is one of the most effective methods to analyze the microstructure of crystal materials [26,27], so, the crystallinity of the modified CD and CD-based acrylamide flocculant were examined by XRD measurements.”

Section 3.1, Structure and morphology, **paragraph 3 (SEM)**, one reference (ref. 28) was added here.

“Figure 3 shows the morphological analyses of the as-received pure β-CD, the TsO-β-CD, EDA-β-CD, AM-β-CD and AM-β-CD-DMDAAC flocculant.” **changed to** “SEM was conducted to characterize the as-received pure β-CD, TsO-β-CD, EDA-β-CD, AM-β-CD and AM-β-CD-DMDAAC flocculant, because it is a highly versatile methodologies for 2D and 3D materials characterization [28]. The representative SEM images are shown in Figure 3.”

Section 3.1, Structure and morphology, **paragraph 4 (TG)**, one reference (ref. 30) was added here.

“The thermal stability of materials is an important factor affecting the application [30],” was added before “Figure 4 shows the thermal stability of TsO-β-CD (a), EDA-β-CD (b), AM-β-CD (c) and AM-β-CD-DMDAAC (d). The initial temperature of weightlessness of TsO-β-CD (curve a) is 190 °C.”

Other references’ order also changed correspondingly in revision.

Q3: The synthesis involves very long hours and this shows the method has no practical applicability due to the high energy cost.

Answer: In the process of our laboratory research, in order to obtain the optimal experimental conditions and yield, we need to strictly control the experimental

parameters, such as reaction temperature, reaction time, etc., which results in a longer analysis time. Actually, if mass production is required, the experimental conditions can be appropriately relaxed to improve the analysis efficiency and reduce energy costs.

Q4: The work reported here lacks novelty. There have been plenty of similar materials reported in literature. The authors need to clearly indicate the novelty of this work before it is considered for publication. According to results it seems to me that this method requires extensive amount of resources which cannot be considered as environmentally friendly.

Answer: The superiority of AM- β -CD-DMDAAC flocculant is mainly in the following aspects: **Firstly**, synthetic flocculant is a hydrophobic and strongly cationic flocculant. **Secondly**, synthetic flocculant is suitable for water-soluble dyes adsorption in a wide pH range (pH 3-7). **Thirdly**, the kinetics of equilibrium is very fast (within 30min). **Fourthly**, adsorption capacity (147.1 mg/g) was higher than other traditional flocculants. **Finally**, in our work and other similar references, many reagents are indeed required for the synthesis and application of flocculants. However, compared with the similar reports, we selected the environmentally friendly β -cyclodextrin as the main material to synthesize a highly efficient hydrophobic composite polymer flocculant, and throughout the course of the study, the amount of harmful substances such as acrylamide has been strictly controlled.

Introduction, “Cyclodextrins are abundant in nature and have biodegradable properties,This project has been designed to synthesize a highly efficient hydrophobic cyclodextrin-based composite polymer flocculant.” **changed to** “ β -cyclodextrins are abundant in nature and have biodegradable and environmentally friendly properties, can be modified by chemical method [10-11], and throughout the course of the study, the amount of harmful reagents used will be strictly controlled to protect the environment.”

Four references (refs. 12, 15, 16 and 17) were added here. **Other references' order also changed correspondingly in revision.**